# Occult hepatitis B virus infection in a Kenyan cohort of HIV infected anti-retroviral therapy naïve adults

**Adil Salyani**[1]*, **Jasmit Shah**[1], **Rodney Adam**[1,2], **George Otieno**[3], **Evelyn Mbugua**[3], **Reena Shah**[1]*

**1** Department of Medicine, Aga Khan University Hospital, Nairobi, Kenya, **2** Department of Pathology, Aga Khan University Hospital, Nairobi, Kenya, **3** Department of Medicine, Kijabe Hospital, Kijabe, Kenya

* adil.suleiman@aku.edu (AS); reena.shah@aku.edu (RS)

**Data Availability Statement:** All relevant data are within the paper and its Supporting Information files.

## Abstract

### Background

Occult hepatitis B virus (HBV) infection (OBI) is a phase of HBV infection characterised by the presence of HBV DNA in the absence of detectable hepatitis B surface antigen (HBsAg). OBI is of concern in the HIV-infected due to high prevalence and risk of HBV reactivation. The prevalence and clinico-demographic characteristics of OBI in anti-retroviral therapy (ART) naïve HIV infected adults in Kenya is unknown.

### Methods

A cross sectional study carried was out at three sites in Kenya. HIV infected ART naïve adults were enrolled and demographic data collected. Blood samples were assayed for HBsAg, HBV DNA, alanine aminotransferase, aspartate aminotransferase, antibodies to hepatitis B surface antigen (anti-HBs) and hepatitis B core antigen (anti-HBc). Data on CD4 count, HIV viral load and platelet count were obtained from medical records.

### Results

Of 208 patients, 199 (95.7%) did not report HBV vaccination, 196 (94.2%) were HBsAg negative, 119 (57.2%) had no HBV markers, 58 (27.9%) had previous HBV infection (anti-HBc positive) and 11 (5.3%) had OBI. All 11 (100%) OBI patients were anti-HBc positive. OBI patients comprised 19.0% of HBsAg negative, anti-HBc positive patients. There was no difference in clinico-demographic characteristics between the overt HBV, OBI and HBV negative patients.

### Conclusion

This was the first study on OBI in ART naïve HIV infected adults in Kenya. The lower OBI prevalence compared to other sub-Saharan African countries could be attributed to lower HBV exposure. Most patients were HBV unexposed and unimmunized, outlining the need to implement guideline recommended immunization strategies.

**Funding:** The author(s) received no specific funding for this work.

**Competing interests:** The authors have declared that no competing interests exist.

## Introduction

Africa faces a big burden of chronic viral diseases including human immunodeficiency virus (HIV) and hepatitis B virus (HBV), which are responsible for substantial morbidity and mortality. Due to shared modes of transmission, HIV-HBV co-infections are common, especially in sub-Saharan Africa (SSA) which accounts for majority of HIV-HBV co-infections worldwide [1]. Co-infection is associated with high liver-related morbidity and mortality [2,3]; hence, all HIV infected persons should undergo HBV screening by testing blood for hepatitis B surface antigen (HBsAg) [4].

The widespread use of nucleic acid testing (NAT) to monitor HBV infection has led to the discovery of an HBsAg negative, HBV DNA positive phase of HBV known as occult hepatitis B virus infection (OBI) [5]. The natural history of OBI in HIV is currently not clear. However, OBI is more prevalent in HIV-infected persons than in HIV-uninfected persons. In addition, OBI is reactivated to HBV in the setting of HIV-induced immunosuppression, use of non HBV-active anti-retroviral therapy (ART) regimens and on withdrawal of HBV-active ART regimens [6,7].

OBI is classified according to HBV antibody profiles as seropositive OBI (anti-HBc and/or anti HBs antibody positive) or seronegative OBI (absence of both anti-HBc and anti-HBs antibodies) [7]. OBI can also be classified according to circulating HBV DNA levels as true and false OBI. True OBI arises from suppression of replication activity and gene expression of covalently closed circular DNA (cccDNA) in the liver leading to low HBV DNA levels in blood (<200 IU/mL) [7]. False OBI on the other hand, follows infection with S-gene escape mutants, which produce an altered HBsAg that cannot be detected by some HBsAg assays, with HBV DNA levels comparable to those in overt HBsAg-positive HBV [8].

The prevalence rate of OBI in HIV-infected patients ranges from 15.1 to 24% in SSA. The prevalence of OBI among HIV infected adults in Kenya, which is endemic for both HIV and HBV, is undefined. Differences in epidemiology, transmission patterns, host factors and genotypes mean results from other regions may not be generalizable to Kenya. The objective of this study was to determine the prevalence and characteristics of OBI in ART naïve HIV-infected adults in Kenya.

## Materials and methods

### Subjects and site

ART naïve HIV-infected adults ≥ 18 years presenting for treatment at three sites in and around Nairobi, Kenya were recruited consecutively until the desired sample size was achieved. Patients who had known chronic hepatitis B infection were excluded. These sites were:

1. Aga Khan University Hospital Nairobi (AKUHN), a tertiary care private university teaching hospital in Nairobi frequented by the urban affluent and health insured patients.

2. AIC Kijabe Hospital (AICKH), a faith based rural hospital located in Kijabe Town, 65 km from Nairobi serving the semi-urban and rural poor.

3. Mbagathi District Hospital (MDH): A government run district hospital in Nairobi serving the urban poor.

Assuming a prevalence of 15.1% as found by Bell et al and Mudawi et al [9,10], using a precision level of 5% and allowing a 10% attrition rate, we arrived at a sample size of 217 patients.

Written informed consent was obtained from all participants and ethical approval was obtained from the research ethics committee at AKUHN for all sites as well as individual institution ethics committees.

### Data collection, sample collection and analysis

A questionnaire capturing demographic data and HBV immunisation status was administered to the patients at all sites. Blood was collected and stored as serum and plasma at -20⁰ Celsius until transport at AKUHN where all testing was carried out per the algorithm in Fig 1.

First, all samples were tested for HBsAg by electrochemiluminescence (ECL) on Cobas e601 Immunoassay Analyser (Hitachi High-Technologies Corporation, Tokyo Japan) using Elecsys HBsAg II immunoassay test kits (Roche Diagnostics, Mannheim, Germany). Samples that tested positive for HBsAg were confirmed on the Abbott Architect i1000sr analyser (Abbott Laboratories, IL, USA) by chemiluminescent microparticle immunoassay (CMIA) using Architect HBsAg Qualitative II assay (Abbott Ireland Diagnostics Division, Sligo, Ireland). Discrepant or indeterminate results were excluded from further analysis. Aspartate aminotransferase (AST) and alanine aminotransferase (ALT) levels were determined on all samples on Cobas c501 (Hitachi High-Technologies Corporation, Tokyo, Japan).

Samples that tested negative for HBsAg were tested for HBV DNA, anti-HBs and anti-HBc. HBV DNA testing was carried out on COBAS AmpliPrep-COBAS TaqMan 48 Analyser (Roche Molecular Systems, Inc., Branchburg, NJ, USA) using COBAS AmpliPrep/COBAS TaqMan HBV Test, version 2.0 (Roche Molecular Systems, Inc., Branchburg, NJ, USA) as per

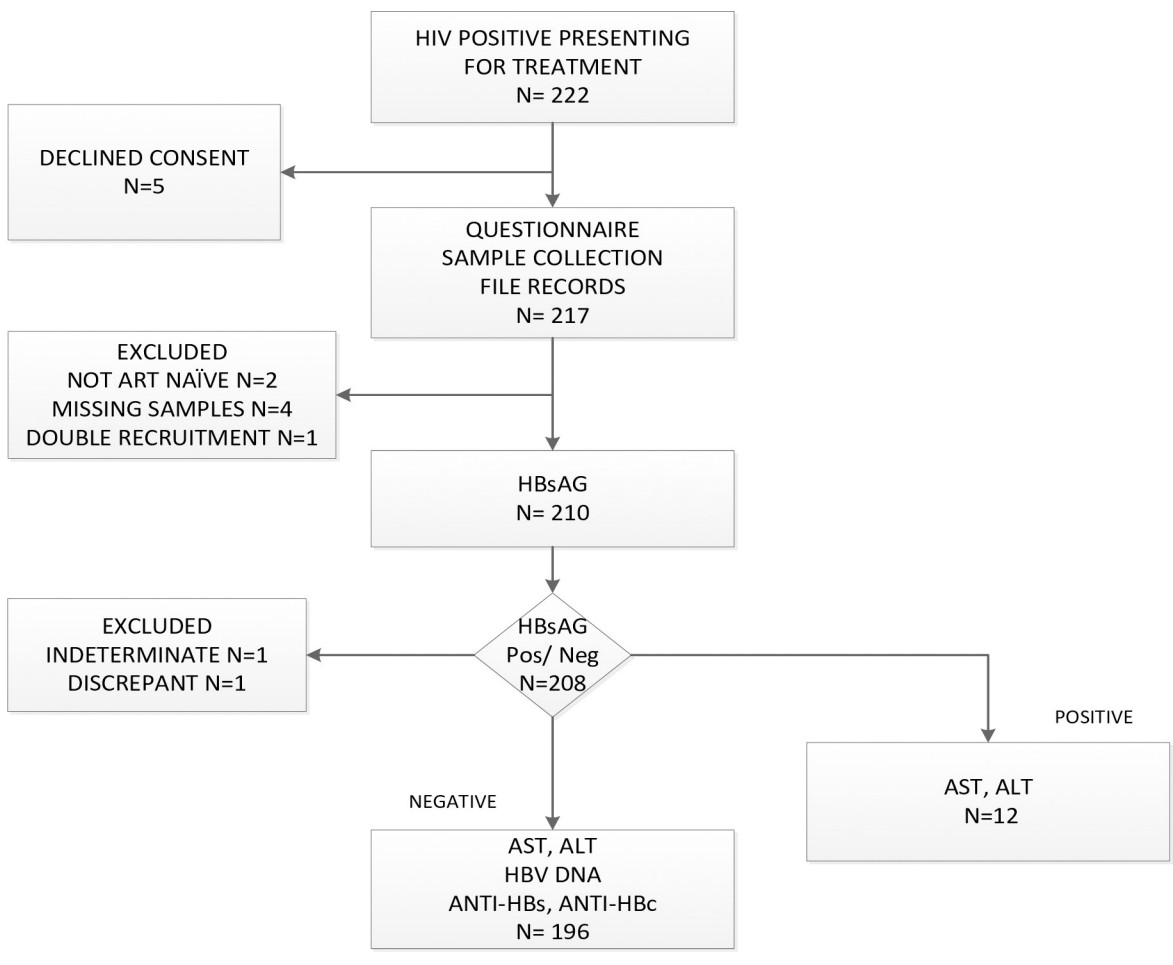

**Fig 1. Study Flow of recruitment and testing.**

manufacturer's instruction using 650 μl of plasma. On this platform, the limit of detection is 10.2 IU/mL [11] and the lower limit for quantitation is $\geq$20 IU/mL.

Testing of anti-HBs and anti HBc on the first 130 samples was done by ECL on Cobas e601 Immunoassay Analyser (Roche Diagnostics, Mannheim, Germany) using Elecsys Anti-HBs II and Elecsys Anti-HBc II kits (Roche Diagnostics, Mannheim, Germany). However due to a change in technology used by the clinical laboratory, the remaining samples tested using an antibody capture CMIA using aHBS2 and HBcT test kits (Siemens Healthcare Diagnostics Inc., Tarrytown NY, USA) on Atellica IM 1600 analyser (Siemens Healthcare Diagnostics Inc., Tarrytown, NY, USA). A validation analysis carried out between these two platforms demonstrated complete concordance of results.

Data on CD4 count, HIV viral load, and platelet count, where available, were obtained from electronic health records at the sites. AST to platelet ratio index (APRI) scores were calculated using the formula: APRI = (AST /AST ULN)/platelet count ($10^9$/L) × 100 [12].

## Statistical analysis

Statistical analysis was carried out on IBM® SPSS® Statistics version 20. Categorical data were presented as frequencies and percentages whereas continuous data were presented as means and standard deviations. Continuous data were then tested for normality using the Shapiro Wilks tests. Univariate analysis was conducted using Chi-squared test or Fisher's exact test for categorical data and Kruskal Wallis test for continuous data.

## Results

Between 22nd March 2019 and 9th April 2020, 222 patients were screened, 217 patients were recruited and data from 208 patients were included in the final analysis (Fig 1). This comprised of 54 patients from AKUHN, 128 from MDH and 26 from AICKH.

### Clinico-demographic characteristics of study population

Of the 208 patients, 113 (54.3%) were female, 108 (51.9%) were married and 199 (95.7%) reported no prior hepatitis B vaccination. Mean age was 38.8 ± 10.6 years (males, 41.3 ± 11.2, females, 36.7 ± 9.7 years) with mean CD4 count and HIV viral load (log 10) of 215 ± 248 cells/μl and 5.22 ± 0.98 copies/ml respectively (Table 1).

### Prevalence of OBI in ART naïve HIV infected patients

From the study group of 208 patients, 12 (5.8%) were hepatitis B surface antigen positive. From the remaining 196 patients, HBV DNA was detected in 11 cases, giving an OBI prevalence of 5.6% among HBsAg negative patients, and 5.3% among the study group (Fig 2). Among the 58 patients who had a resolved natural infection (HBsAg negative, anti-HBc positive), 11 (19.0%) had OBI.

### Prevalence of true and false OBI

Of the 11 OBI cases, 10 (90.9%) had true OBI (HBV DNA <200 IU/mL). Of the true OBI patients, seven (70.0%) had detectable but unquantifiable HBV DNA (<20 IU/mL) while the other three had HBV DNA levels between 43–108 IU/mL. The one case of false OBI had HBV DNA levels of 322 IU/mL.

**Table 1. Comparison of clinico-demographic characteristics of OBI to overt HBV and HBV negative.**

| Parameter | All patients (N = 208)* | Overt HBV (N = 12)* | OBI (N = 11)* | HBV Negative (N = 185)* | P Value° |
|---|---|---|---|---|---|
| Age, years | 38.78 (10.63) | 42.08 (13.03) | 45.73 (13.73) | 38.15 (10.13) | 0.144 |
| Female sex, n (%) | 113 (54.3%) | 5 (41.7%) | 3 (27.7%) | 105 (56.8%) | 0.107 |
| Marital Status, n (%) | | | | | 0.439 |
| Single | 55 (26.4%) | 1 (8.3%) | 1 (9.1%) | 53 (28.7%) | |
| Married | 108 (51.9%) | 9 (75.0%) | 8 (72.7%) | 91 (49.2%) | |
| Separated | 13 (6.3%) | 0 (0.0%) | 1 (9.1%) | 12 (6.5%) | |
| Divorced | 16 (7.7%) | 1 (8.3%) | 0 (0.0%) | 15 (8.1%) | |
| Widowed | 16 (7.7%) | 1 (8.3%) | 1 (9.1%) | 14 (7.6%) | |
| No HBV vaccination, n (%) | 199 (95.7) | 12 (100.0) | 10 (90.9) | 177 (95.7) | 0.514 |
| CD4, cells/μL | 215.30 (248.05) | 62.50 (41.72) | 201.60 (145.09) | 219.71 (255.05) | 0.696 |
| HIV VL, $\log_{10}$ copies/mL | 5.22 (0.98) | 5.43 (.) | 4.74 (1.14) | 5.25 (0.98) | 0.574 |
| AST, IU/L | 28.85 (28.57) | 53.38 (92.27) | 37.39 (16.88) | 26.73 (16.84) | 0.064 |
| ALT, IU/L | 14.52 (15.90) | 21.89 (37.60) | 21.65 (18.83) | 13.61 (13.13) | 0.257 |
| Platelets, cells $10^9$/L, | 259.83 (89.55) | 146.00 (.) | 198.75 (90.26) | 266.46 (88.00) | 0.130 |
| APRI score | 0.36 (0.30) | 0.61 (.) | 0.85 (0.85) | 0.31 (0.19) | 0.054 |

Data represents mean (standard deviation in parentheses), except when indicated.

* N was not the same down the column because variable availability of results.

° Calculated with Kruskal- Wallis 1-way Anova.

### Serologic profile of patients with OBI

All 11 (100%) of OBI patients were seropositive (anti-HBc and/or anti HBs positive) with nine (81.8%) of them having both anti-HBc and anti-HBs while two (18.2%) had an isolated anti-HBc. None of the OBI patients had isolated anti-HBs or was seronegative. There were significant differences in antibody profiles of patients with OBI compared to HBV DNA negative patients. While anti-HBc was positive in all 11 (100%) OBI patients, only 47 of 185 (25.4%) of HBV DNA negative patients had detectable anti-HBc, P <0.001 (Table 2).

### Comparison of OBI patients to overt HBV and HBV negative

The clinico-demographic characteristics of these patients are compared in Table 1. Although there were trends where patients with overt HBV and OBI were older, male and had higher AST levels and APRI score, none of them reached significance.

### Exposure to HBV in the population

HBV antibody markers were only done on patients who were negative for HBsAg. Of the 196 HBsAg negative patients, 119 (60.7%) were seronegative to anti-HBc and anti-HBs suggesting no previous HBV exposure, 58 (29.6%) were anti-HBc positive with or without anti-HBs suggesting resolved past infection and 19 (9.7%) had isolated anti-HBs suggesting previous vaccination. Therefore, from the whole population of 208, 119 (57.2%) had no HBV markers and therefore susceptible to infection.

Comparing the demographic characteristics of those with any marker of past or current HBV infection (HBsAg, HBV DNA or anti-HBc) to those devoid of all HBV markers (HBV unexposed), showed that the HBV unexposed group were likely to be younger and never married compared to those with a current/past HBV infection (P 0.012 and 0.007, respectively) (Table 3).

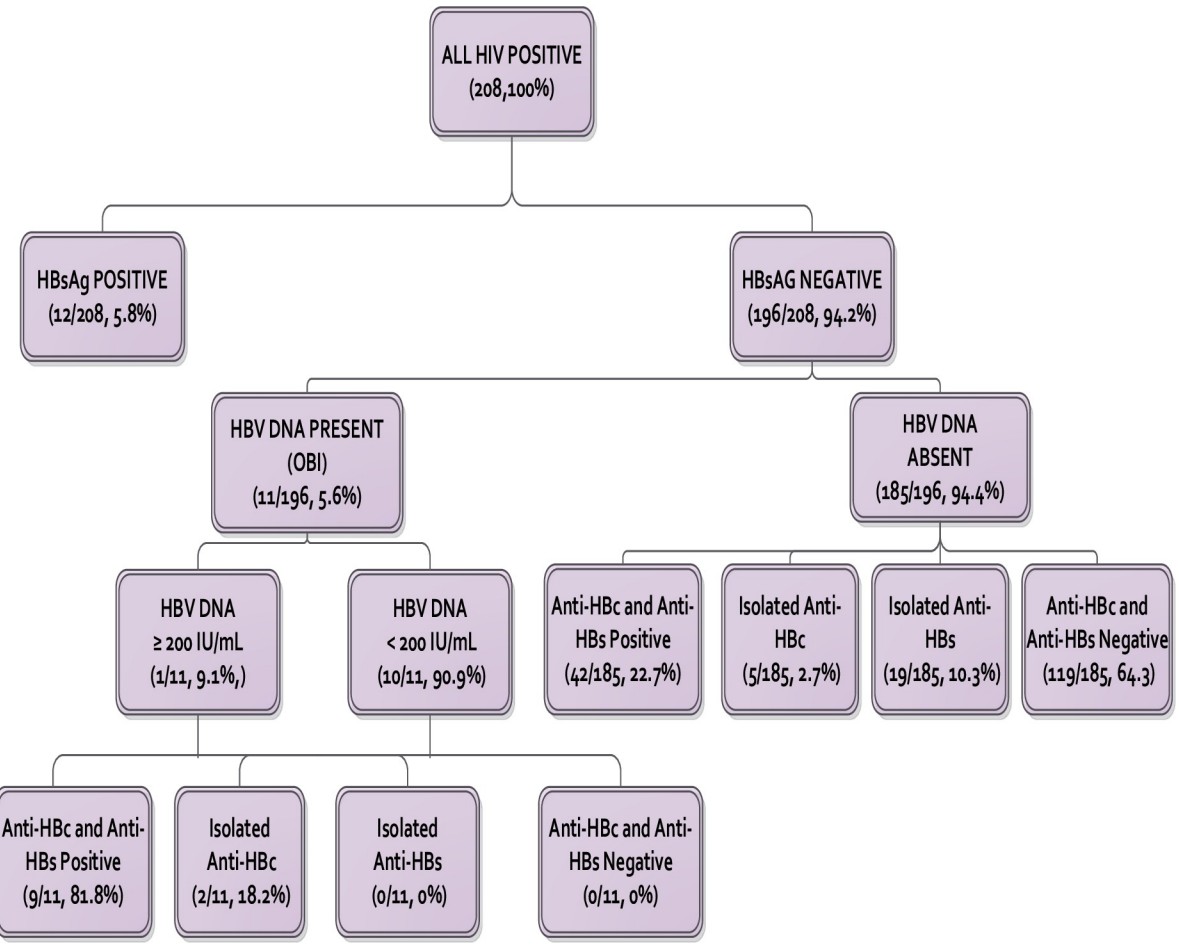

**Fig 2. HBV serological and DNA markers.**

## HBV vaccination

Of the 19 patients with a profile suggesting previous vaccination, only five patients (26.3%) reported vaccination. Conversely, of the nine patients who reported vaccination, only four (44.4%) had a primary anti-HBs response (anti HBs titre of <10 IU/L after a single dose and >10 IU/L after a three doses). One patient had titres of <10 IU/L despite three doses of the vaccine while three patients (15.8%) had no detectable antibodies, suggesting a misreporting

**Table 2. Antibody profiles of OBI and HBV DNA negative patients.**

| Antibody profile | OBI (N = 11) N, (%) | No HBV (N = 185) N, (%) | P value° |
|---|---|---|---|
| Anti-HBc +, anti-HBs + | 9 (81.8) | 42 (22.7) | **<0.001** |
| Anti-HBc+, anti-HBs - | 2 (18.2) | 5 (2.7) | 0.052 |
| Anti-HBc -, anti-HBs + | 0 (0.0) | 19 (10.27) | 0.393 |
| Anti-HBc -, anti HBs - | 0 (0.0) | 119 (64.3) | **<0.001** |
| Anti-HBc + | 11 (100.0) | 47 (25.4) | **<0.001** |
| Anti-HBs + | 9 (81.8) | 61 (33.0) | **0.002** |

° Calculated with Fisher Exact Probability Test.

**Table 3. Comparison of demographic characteristics of current/past HBV infection to HBV unexposed.**

| Parameter | Total (N = 189) | Current/ past HBV infection (N = 70) | HBV unexposed (N = 119) | P Value˚ |
|---|---|---|---|---|
| Age in years, mean (SD) | 38.4 (10.5) | 41.4 (12.4) | 36.6 (8.9) | 0.012 |
| Female sex, number (%) | 101 (53.4%) | 32 (45.7) | 69 (58.0) | 0.102 |
| Marital Status, number (%) | | | | 0.007 |
| Married/Divorced/Separated/Widowed | 141 (74.6) | 60 (85.7) | 81 (68.1) | |
| Single | 48 (25.4) | 10 (14.3) | 38 (31.9) | |

˚ Calculated with Kruskal- Wallis 1-way Anova.

or unsuccessful vaccination. One patient who had received three doses of the vaccination, was both anti-HBs and anti-HBc positive suggesting a natural infection preceding the vaccination. This patient also had detectable HBV DNA in the range for a true OBI.

## Discussion

In this cohort of 208 ART naïve patients recruited from hospitals serving patients from varying socio-economic backgrounds, we found an HBsAg prevalence of 5.8%. Although the protocol of this study was not designed to estimate the prevalence of HBsAg among ART naïve patients, the presence of known chronic HBV infection being an exclusion criterion, no patient was actually excluded due to this criterion (Fig 1). The findings of this study with that regard therefore remain valid.

This HBsAg prevalence was similar to the 6% found by Harania in 2008 at AKUHN, one of the sites in this present study [13] but lower than what is described in other parts of SSA of 8.7 to 11.7% [9,10,14–16]. This is in keeping with Kenya being one of the few countries with intermediate HBV endemicity in a region that is generally hyper-endemic for HBV [17].

The prevalence of HBV-HIV coinfection in adults remain largely unchanged since the last study in 2008, above [13]. The benefits of universal infant vaccination for Hepatitis B, introduced slightly over 18 years ago in November 2001 as part of the Kenya Expanded Programme on Immunization (EPI) [18], would not be apparent in our cohort as the youngest patient was 19 years old at enrolment and therefore born before this came in place. This is however is likely to change in the coming decades.

The prevalence of OBI in our cohort was 5.6% of those without HBsAg and 5.3% of all the HIV patients. This was lower than the 18.7% described by Jepkemei et al in a Kenyan study of a heterogeneous population of individuals at high risk of HBV infection, some of whom were HIV positive and on anti-retroviral therapy. This was mainly driven by the high prevalence among patients with symptomatic liver disease, who made up more than a third of the study population [19]. This prevalence is also much lower than what is described in other sub-Saharan countries where it ranges from 15.1 to 24% among the HIV infected [9,10,14,20]. Since OBI prevalence mirrors that of HBsAg and HBV exposure [7], the lower prevalence of OBI in Kenya is commensurate with the lower prevalence of these indices in Kenya [17]. This finding was also apparent in this study for both HBsAg as described above and HBV exposure, where only 32.1% of patients from our cohort had evidence of current/past HBV infection as compared to 62.8% to 77.5% in the comparative studies from SSA [9,10,20]. In adjusting for the lower HBV exposure by considering only those who had evidence of a previous HBV infection, nearly one in five (19%) of patients with resolved HBV had OBI.

The prevalence of false OBI was low in our cohort with only one of the 11 OBI patients meeting that definition. There was a similarly low prevalence of 8.3% seen in Botswana [14] but much higher prevalence in the region of 32.4% to 93.3% are reported from Sudan and

South Africa [9,10,20]. The prevalence of false OBI would reflect that of S-gene escape mutants. Based on these findings, we would expect a low level of S-gene escape mutants in Kenya, a finding seen in among healthy blood donors where only 2 of 69 (2.9%) of HBV DNA positive donors had undetectable HBsAg [21]. Both of these individuals harboured the T143M mutation that is associated with false negative HBsAg assays and viral escape [22]. However, these mutations were more common in OBI patients with symptomatic liver disease, where they were detected in 3 of 13 patients (23.1%) [19].

There were no seronegative cases of OBI in our cohort and all the cases of OBI were anti-HBc positive with the majority of them (81.8%) being positive for anti-HBc and anti-HBs. These findings are different from what is described elsewhere in SSA with seronegative OBI rates in HIV infected of 26.7% to 30% [9,14,16,20]. Seronegative OBI arises due to either progressive loss of antibodies or is antibody negative from onset of infection [7] and poses a challenge to the use of anti-HBc testing to identify OBI patients. The absence of seronegative OBI in our cohort would therefore allow anti-HBc to be used to identify OBI patients.

The effect of OBI on liver fibrosis and cirrhosis is still unknown. Bell et al in South Africa and Carimo et al in Mozambique found higher APRI scores in OBI patients compared to HBV negative patients, [9,15] a finding not demonstrated by Ryan et al in Botswana [14]. Our study, which was limited by low number of patients who had these scores available, showed numerically higher APRI scores in overt and occult HBV infection compared to HBV negative, although this did not reach significance (P 0.054). Further studies with higher numbers are needed to shed more light on this.

Vaccination is recommended for all HBV non-immune HIV patients by the World Health Organization (WHO) [23]. The American Association for the Study of Liver Diseases (AASLD) recommends also vaccinating HIV patients with isolated anti-HBc as they are still susceptible to HBV infection due to lack of protective antibodies. The majority of our cohort (57.2%) had no HBV markers; hence were susceptible to infection and candidates for vaccination. The rates of HBV vaccination of HIV patients are not well documented in Kenya but are expected to be low as inferred from the study by Harania at AKUHN where 81.7% of patients with HIV were not vaccinated for HBV [13]. Unlike ART, which is provided free, HBV vaccination needs to be paid for by patients in Kenya, which may explain the low uptake.

The use of HBV-active ART (tenofovir, lamivudine and emtricitabine) has shown to reduce the incidence of new HBV infection in HIV patients [24,25]. Kenya has adopted the WHO guidelines for treatment of HIV; these have either tenofovir or lamivudine or both as part of all the preferred first line ART for adults in the both the 2016 and 2019 updates [23,26], therefore offering some protection from HBV to the unvaccinated patients. However, the use of HBV-active ART is not considered a preferred strategy to prevent HBV infection in HIV due to lack of long-term immunity afforded by this method [27] as well as infection by drug resistant strains [28]. Poor compliance to ART would also be expected to deleteriously affect the efficacy of such a strategy.

Self-report of HBV vaccine was not a reliable marker of HBV immunity in our cohort with only four of the nine patients with a history of vaccination showing a primary vaccine response. This could have arisen from either misreporting or vaccine non-responsiveness, which occurs in a high proportion in HIV infected patients [29,30]. In addition, just over a quarter of patients who had a serological profile consistent with vaccination actually reported vaccination. Therefore, in this setting, it would be prudent to use anti-HBs titres rather than patient history in assessing need for vaccination.

In conclusion, our study, the first on OBI in ART naïve HIV infected adults in Kenya showed a lower prevalence of OBI, HBsAg and HBV exposure compared to other SSA countries, in keeping with Kenya as a region of intermediate HBV endemicity. All OBI patients had

positive anti-HBc, making it a useful marker for identifying OBI. We also found HBV immunisation history not to be a reliable marker of immunity to HBV, emphasising the importance of laboratory testing for this purpose.

## Supporting information

**S1 File. Data set OBI-Kenya.**
(XLSX)

## Acknowledgments

We thank Dr Sayed Karar for proofreading and editing of this paper, Prof. Zahir Moloo and James Ndungu for availing the laboratory facilities at AKUHN and Dr Geoffrey Omuse, Dr Daniel Maina, Dr Abubakar Abdillah and Dr Enock Serem for their invaluable advice on all matters laboratory related. We would like to appreciate the support of Dr Nancy Abuya of Mbagathi District Hospital towards this study at that site. We are indebted to Josephine Wanja Mburu, Tahera Khatau, Susan Karimi, Rose Njenga, Ephantus Mbugua, Boniface Shitambasi, Aidah Muthui, Paul Sigey, Jacob Mosima, Maxwell Odhiambo, Joyce Njoroge, Edith Waweru and team and Mr David Macharia for their sterling work in screening and recruiting patients as well as sample collection, storage and analysis.

## Author Contributions

**Conceptualization:** Adil Salyani, Rodney Adam, Reena Shah.

**Data curation:** Adil Salyani, George Otieno, Evelyn Mbugua.

**Formal analysis:** Adil Salyani, Jasmit Shah, Rodney Adam.

**Investigation:** Adil Salyani, George Otieno, Evelyn Mbugua.

**Methodology:** Adil Salyani, Jasmit Shah, Rodney Adam, Reena Shah.

**Project administration:** Reena Shah.

**Resources:** Rodney Adam, George Otieno, Evelyn Mbugua, Reena Shah.

**Supervision:** Jasmit Shah, Rodney Adam, Reena Shah.

**Validation:** Adil Salyani, Jasmit Shah, Rodney Adam, George Otieno, Evelyn Mbugua, Reena Shah.

**Visualization:** Adil Salyani, Jasmit Shah, Rodney Adam, Reena Shah.

**Writing – original draft:** Adil Salyani.

**Writing – review & editing:** Adil Salyani, Jasmit Shah, Rodney Adam, Reena Shah.

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
