## [Decision Letter · Decision Letter 0]

5 Nov 2020

PONE-D-20-32638

Occult hepatitis B virus infection in a Kenyan cohort of HIV infected anti-retroviral therapy naïve adults

PLOS ONE

Dear Dr. Salyani,

Thank you for submitting your manuscript to PLOS ONE. After careful consideration, we feel that it has merit but does not fully meet PLOS ONE’s publication criteria as it currently stands. Therefore, we invite you to submit a revised version of the manuscript that addresses the points raised during the review process. Some are details but of importance, precisions, some may require experimental testing such as HBV genotyping or mutant's presence.

We look forward to receiving your revised manuscript.

Kind regards,

Isabelle Chemin, PhD

Academic Editor

PLOS ONE

Journal Requirements:

Reviewers' comments:

Reviewer's Responses to Questions

**Comments to the Author**

1. Is the manuscript technically sound, and do the data support the conclusions?

Reviewer #1: Yes

2. Has the statistical analysis been performed appropriately and rigorously? 

Reviewer #1: Yes

3. Have the authors made all data underlying the findings in their manuscript fully available?

Reviewer #1: Yes

4. Is the manuscript presented in an intelligible fashion and written in standard English?

Reviewer #1: Yes

5. Review Comments to the Author

Reviewer #1: This is a cross-sectional study of occult HBV infection in ART-naïve HIV positive individuals in Kenya. Given the high burden of HBV in resource-limited settings, studies such as this are important.

Overall, the study population size is modest at 208 individuals and the findings are not surprising based on similar studies in other sub-Saharan African countries.

Storing serum / plasma at -20C is never the best option. Storage at -80C is better.

Were HBV DNA negative samples re-tested a second time? This would increase the rigor of the study design.

Was HBV DNA quantified in duplicate or triplicate? Again, this would increase the rigor of the study design.

Given that only 11 individuals tested positive for occult HBV infection, the authors should evaluate the HBV genotypes and presence/absence of occult-associated mutations as well.

For reference 14, the prevalence of occult HBV was not 8.3% as stated in lines 221-223 of the Discussion. Rather, HBV DNA was detected in 72 of 272 (26.5%) who previously tested HBsAg negative.

The recent study by Jepkemei et al. about occult HBV in Kenya should be referenced and considered in the Discussion.

Characterization of occult hepatitis B in high-risk populations in Kenya. Jepkemei KB, Ochwoto M, Swidinsky K, Day J, Gebrebrhan H, McKinnon LR, Andonov A, Oyugi J, Kimani J, Gachara G, Songok EM, Osiowy C. PLoS One. 2020 May 28;15(5):e0233727.

6. PLOS authors have the option to publish the peer review history of their article (what does this mean?). If published, this will include your full peer review and any attached files.

Reviewer #1: No

---

## [Author Response · Author response to Decision Letter 0]

6 Dec 2020

Comment 1: This is a cross-sectional study of occult HBV infection in ART-naïve HIV positive individuals in Kenya. Given the high burden of HBV in resource-limited settings, studies such as this are important.

Response 1: We agree that since both HIV and HBV are endemic to Kenya and due to the clinical implications of occult HBV in HIV patients, this study was important to carry out.

Comment 2: Overall, the study population size is modest at 208 individuals and the findings are not surprising based on similar studies in other sub-Saharan African countries.

Response 2: The sample size was calculated based on an estimated prevalence of 15.1% and precision value of 5%. Our findings revealed a lower OBI prevalence compared to other sub-Saharan countries, which we attribute to a lower exposure to HBV in the population studied, also a key finding of our study. We also found that all the OBI patients in our study had positive anti-HBc, which would be a useful marker for identifying OBI in this population. Moreover, we explored the relationship between a history of HBV vaccination and immunity and we found that a history of vaccination was not a reliable marker of immunity in this population. All these aspects make our study unique from those done previously in sub-Saharan Africa.

Comment 3: Storing serum / plasma at -20C is never the best option. Storage at -80C is better.

Response 3: We would note that as a DNA virus, Hepatitis B is stable for an extended amount of time at -200 C. A study on stability of HBV nucleic acids in plasma samples after long term storage demonstrated a loss of less than 0.5 log10 in HBV DNA after storage for up to 5 years at -200 C [1]. All our samples underwent analysis within a year of collection; we therefore believe that -200 C storage was appropriate for our study.

Comment 4: Were HBV DNA negative samples re-tested a second time? This would increase the rigor of the study design. Was HBV DNA quantified in duplicate or triplicate? Again, this would increase the rigor of the study design.

Response 4: The samples were run on the Roche COBAS AmpliPrep- COBAS TaqMan 48 Analyser, which has full internal controls for every sample. In addition, the laboratory is using the same platform for clinical testing. As a College of American Pathologists (CAP) accredited laboratory, regular external quality assessment is performed which includes Calibration Verification/Linearity (CVL). The laboratory has passed all these satisfactorily. We respectfully disagree with the necessity of repeat testing in this setting.

Comment 5: Given that only 11 individuals tested positive for occult HBV infection, the authors should evaluate the HBV genotypes and presence/absence of occult-associated mutations as well.

Response 5: We agree that this would have been useful information. However, the determination of genotype is beyond the scope and budget of this study, so we were not able to perform this.

Comment 6: For reference 14, the prevalence of occult HBV was not 8.3% as stated in lines 221-223 of the Discussion. Rather, HBV DNA was detected in 72 of 272 (26.5%) who previously tested HBsAg negative.

Response 6: The discussion in this instance was about the prevalence of FALSE OBI, not all cases of OBI. The prevalence of false OBI in the study by Ryan in Botswana was 8.3% as we have stated in the manuscript.

Comment 7: The recent study by Jepkemei et al. about occult HBV in Kenya should be referenced and considered in the Discussion.

Characterization of occult hepatitis B in high-risk populations in Kenya. Jepkemei KB, Ochwoto M, Swidinsky K, Day J, Gebrebrhan H, McKinnon LR, Andonov A, Oyugi J, Kimani J, Gachara G, Songok EM, Osiowy C. PLoS One. 2020 May 28;15(5):e0233727.

Response 7: We thank you for bringing this study to our attention. We have included it in the discussion and referenced it as appropriate. We have also edited our manuscript to reflect that our study is not the first on OBI in Kenya as earlier stated, but the first in ART naïve HIV infected adults.

---

## [Decision Letter · Decision Letter 1]

21 Dec 2020

Occult hepatitis B virus infection in a Kenyan cohort of HIV infected anti-retroviral therapy naïve adults

PONE-D-20-32638R1

Dear Dr. Salyani,

We’re pleased to inform you that your manuscript has been judged scientifically suitable for publication and will be formally accepted for publication once it meets all outstanding technical requirements.

Kind regards,

Isabelle Chemin, PhD

Academic Editor

PLOS ONE

Additional Editor Comments (optional):

Reviewers' comments:

Reviewer's Responses to Questions

**Comments to the Author**

1. If the authors have adequately addressed your comments raised in a previous round of review and you feel that this manuscript is now acceptable for publication, you may indicate that here to bypass the “Comments to the Author” section, enter your conflict of interest statement in the “Confidential to Editor” section, and submit your "Accept" recommendation.

Reviewer #1: All comments have been addressed

2. Is the manuscript technically sound, and do the data support the conclusions?

Reviewer #1: Yes

3. Has the statistical analysis been performed appropriately and rigorously? 

Reviewer #1: Yes

4. Have the authors made all data underlying the findings in their manuscript fully available?

Reviewer #1: Yes

5. Is the manuscript presented in an intelligible fashion and written in standard English?

Reviewer #1: Yes

6. Review Comments to the Author

Reviewer #1: No additional comments / all previous comments have been addressed adequately.

7. PLOS authors have the option to publish the peer review history of their article (what does this mean?). If published, this will include your full peer review and any attached files.

Reviewer #1: No

---

## [Editor Report · Acceptance letter]

28 Dec 2020

PONE-D-20-32638R1 

Occult hepatitis B virus infection in a Kenyan cohort of HIV infected anti-retroviral therapy naïve adults. 

Dear Dr. Salyani:

I'm pleased to inform you that your manuscript has been deemed suitable for publication in PLOS ONE. Congratulations! Your manuscript is now with our production department. 

Kind regards, 

on behalf of

Mrs Isabelle Chemin 

Academic Editor

PLOS ONE